# Real-Time Postural Disturbance Detection Through Sensor Fusion of EEG and Motion Data Using Machine Learning

**DOI:** 10.3390/s24237779

**Published:** 2024-12-05

**Authors:** Zhuo Wang, Avia Noah, Valentina Graci, Emily A. Keshner, Madeline Griffith, Thomas Seacrist, John Burns, Ohad Gal, Allon Guez

**Affiliations:** 1Department of Electrical and Computer Engineering, Drexel University, Philadelphia, PA 19104, USA; guezal@drexel.edu; 2Center for Injury Research and Prevention, Children’s Hospital of Philadelphia, Philadelphia, PA 19104, USA; graciv@chop.edu (V.G.); griffithm1@chop.edu (M.G.); seacrist@chop.edu (T.S.); johnjburns21@gmail.com (J.B.III); 3GraceFall, Inc., Penn Valley, PA 19702, USA; avian@ruppin.ac.il (A.N.); emily.keshner@temple.edu (E.A.K.); udigal@yahoo.com (O.G.); 4Faculty of Engineering, Ruppin Academic Center, Emek-Hefer, Monash 40250, Israel; 5School of Biomedical Engineering, Science, and Health System, Drexel University, Philadelphia, PA 19104, USA

**Keywords:** fall detection, EEG, system identification, elderly adults, sensor fusion

## Abstract

Millions of people around the globe are impacted by falls annually, making it a significant public health concern. Falls are particularly challenging to detect in real time, as they often occur suddenly and with little warning, highlighting the need for innovative detection methods. This study aimed to assist in the advancement of an accurate and efficient fall detection system using electroencephalogram (EEG) data to recognize the reaction to a postural disturbance. We employed a state-space-based system identification approach to extract features from EEG signals indicative of reactions to postural perturbations and compared its performance with those of traditional autoregressive (AR) and Shannon entropy (SE) methods. Using EEG epochs starting from 80 ms after the onset of the event yielded improved performance compared with epochs that started from the onset. The classifier trained on the EEG data achieved promising results, with a sensitivity of up to 90.9%, a specificity of up to 97.3%, and an accuracy of up to 95.2%. Additionally, a real-time algorithm was developed to integrate the EEG and accelerometer data, which enabled accurate fall detection in under 400 ms and achieved an over 99% accuracy in detecting unexpected falls. This research highlights the potential of using EEG data in conjunction with other sensors for developing more accurate and efficient fall detection systems, which can improve the safety and quality of life for elderly adults and other vulnerable individuals.

## 1. Background and Aim

Falls are a major public health problem worldwide. Falls can result in injuries such as fractures, head injuries, and lacerations, which can lead to hospitalization, disability, and even death [1]. According to the World Health Organization [2], an estimated 646,000 falls result in death globally each year, making it the second-leading cause of unintentional injury death after road traffic accidents. Falls also result in non-fatal injuries, with an estimated 37.3 million severe falls requiring medical attention each year [2]. The financial burden of falls is also significant. In the United States alone, falls result in approximately USD 50 billion in direct medical costs annually [3], and this figure is expected to rise to over USD 101 billion with an aging population by the year 2030 [4]. Falls also have indirect costs, including lost productivity, reduced quality of life, and caregiver burden [5].

Elderly adults have an increased likelihood of falls due to age-related changes in physical and cognitive function [6]. In the United States, it is estimated that approximately 27–30% of people aged 65 and elderly fell annually over the past ten years [7]. In addition to the elderly population, other groups that are prone to falling include adults with neurological disorders, such as Parkinson’s disease, Alzheimer’s disease, and epilepsy seizures, and individuals with visual impairments and athletes. With up to 70% of people with Parkinson’s experiencing at least one fall per year, the leading causes of falls in Parkinson’s disease include gait and balance disturbances, freezing of gait, and medication side effects [8]. People experiencing an epilepsy seizure are more likely to experience a loss of awareness and muscle control, which may cause a collapse to the ground [9]. Similarly, studies reported that individuals with visual impairments are three to four times more likely to experience falls than those without visual impairments. Factors that contribute to falls in this population include decreased visual acuity, contrast sensitivity, and depth perception [10]. Finally, otherwise healthy athletes in sports such as gymnastics, ice skating, and snowboarding are at risk of falls and associated injuries. In these sports, falls can occur due to factors such as muscle imbalances, core stability deficits, and poor neuromuscular control [11].

Fortunately, with the advancement of technologies, the number of proposed fall detection systems and devices has increased dramatically. Wu et al. [12] developed a fall detection system that collected acceleration and rotation angle information from a subjects’ waist and identified a threshold for the sum of acceleration that would detect falls. The system proposed in [13] utilized the accelerometer of a smartphone and incorporated GPS-based real-time location tracking for prompt emergency intervention. These innovations and similar devices rely on inertial sensors to trigger fall detection, allowing for the identification of key time points in the motion of a fall, from onset to completion. Despite these advancements, distinguishing between an actual loss of stability and non-fall activities remains challenging, leading to a high false alarm rate.

Attempts have been made to increase a fall detection device’s reliability through sensor fusion systems with cameras, radar, and acoustic systems. For instance, ref. [14] utilized artificial vision methods to monitor the presence of individuals in a room and detect falls by analyzing changes in their posture or lack of movement. Radar technologies were also used in studies [15] to detect falls by monitoring a change in motion frequencies. The authors of [16] proposed an acoustic fall detection system that utilized a circular microphone array to capture and analyze sounds for automatic fall detection. However, these methods are constrained by the active space within the location where the systems are installed. Moreover, the challenge of distinguishing real falls from activities of daily living (ADLs) persists [17], and this poses a challenge in finding a practical threshold between the missed fall detection rate and high false alarm rate, rendering these devices inefficient.

Recent research [18,19] suggested that wearable EEG devices may provide a solution to the challenge of accurately detecting real falls. An electroencephalogram, or EEG, is a noninvasive method of measuring brain activity that is commonly used in applications such as epilepsy seizure and sleep monitoring [20]. One type of cortical activation measured by EEG is the event-related potential (ERP), which can be elicited with balance perturbations [21]. This brain activity, known as perturbation-evoked potentials (PEPs), can be identified through a single electrode site [22], and are widely distributed over the frontal, central, and parietal cortexes. Additionally, correlation analysis demonstrated a good match between mastoid and cortical EEGs through well-known EEG paradigms [23]. Furthermore, advancements such as the development of dry-ear EEG sensors with flexible earpieces [24] demonstrate the potential for more convenient and cost-effective brain–computer interface applications. For practical application in real-world scenarios, integrating EEG signals with acceleration data from motion sensors enables earlier and more accurate fall prediction compared with existing solutions, thereby helping to prevent injuries.

Based on these findings, we previously conducted an experiment to differentiate PEPs into predictable and unpredictable postural disturbances using EEG recordings in young and elderly adults [25]. The Root-Mean-Square (RMS) values of EEG epochs from cortical and mastoid channels were analyzed using the three-way Repeated Measures Analysis of Variance (ANOVA) to evaluate the effects of the perturbation type. Our results show that the predictability of a postural disturbance modified the EEG activity related to balance adjustments and that this EEG alteration could be identified using a single electrode site placed over the mastoid. It is noteworthy that the labeling method used in our previous study was based on the index and direction of the perturbations rather than the EEG activity.

In the current study, we expanded upon our previous research [25] by implementing a supervised manual data labeling method that categorized the disturbances based on the EEG responses rather than the perturbation index or direction. This labeling method provided a cortical-activity-focused perspective on balance adjustments, potentially enhancing the detection of events that may have been overlooked by the previous labeling method.

A state-space-based system identification method was applied to evaluate the impact of varying EEG epoch lengths based on the classification performance. This approach was proved effective in prior studies, including sleep stage classification [26] and seizure detection [27]. Its efficacy was further demonstrated through comparisons with widely used approaches, such as autoregressive (AR) modeling, Shannon entropy (SE), and support vector machine (SVM) classifiers [28,29,30,31]. In addition, a real-time algorithm integrating EEG and accelerometer data underscores the system’s potential for rapid and accurate fall events classification, offering practical advantages for safety applications.

The objective of this study was to develop and evaluate a novel approach for real-time fall detection through the supervised classification of PEPs from ongoing mastoid EEG recordings. Our proposed method incorporates advanced feature extraction and machine learning techniques, as well as a comprehensive evaluation of various epoch lengths. Additionally, a real-time detection algorithm is introduced, enabling the rapid and accurate classification of fall events while allowing sufficient time for communication and protective measures, such as airbag deployment. By improving the accuracy and reliability of fall detection systems, particularly in distinguishing between actual falls and activities of daily living (ADLs), this work ultimately aimed to support the development of fall detection technologies that are effective and accessible to the elderly and other vulnerable populations, thereby enhancing their safety and overall quality of life.

## 2. Method

The protocol of this study was reviewed and approved by the Institutional Review Board of the Children’s Hospital of Philadelphia, and the participants were compensated for their participation.

### 2.1. Participants

Recruitment was conducted through ResearchMatch.org, a platform that connects individuals with research studies based on their health profiles. This study included 40 participants, comprising 20 young adults (mean age: 24.6 ± 5.9 years; height: 172.2 ± 6.9 cm; weight: 69.6 ± 10.6 kg) and 20 older adults (mean age: 66.9 ± 5.8 years; height: 169.4 ± 8.4 cm; weight: 68.8 ± 10.3 kg). Each group was evenly divided by gender, with 10 males and 10 females.

The exclusion criteria included a weight exceeding 195 lbs; cognitive impairments; use of walking aids; a history of spine, pelvis, or lower extremity fractures within the past five years; pregnancy; uncorrected vision or hearing issues; significant foot deformities or amputations; hip or knee replacements; peripheral neuropathy; medications affecting postural control; and a fear of falling or amusement park rides.

### 2.2. Procedures

All subjects wore a safety harness and were asked to maintain an upright stance on a dynamic platform measuring 60 cm by 90 cm, with their feet placed shoulder-width apart and eyes open (Figure 1A). The experimental protocol called for each subject to experience 34 balance perturbations, consisting of 29 posterior and 5 anterior translations of the platform of 20 cm displacement at a rate of 100 cm/s and acceleration of 0.2 g, for a total of 1360 translations. However, due to technical or subject-related issues, 40 translations across the group were not executed, yielding a total of 1320 completed translations.

During the experiment, active electrodes over an EasyCap (Brain Vision, Morrisville, NC, USA) were located according to the international 10–20 system at positions Fz, Cz, F7, F8, C3, C4, P7, P8, T7, and T8 (Figure 1B). Additionally, 3 channels from cEEGrids (TMSI, Oldenzaal, Netherlands) were patched over each mastoid at L5, L6, L7, R5, R6, and R7 (Figure 1C–E). Reference electrodes were placed at FCz and AFz. Raw EEG data were sampled at 1000 Hz and filtered offline with a 2nd-order Butterworth bandpass filter (2.5 Hz–30 Hz) and a 60 Hz notch filter. The platform acceleration was sampled at 150 Hz (Delsys, Inc., Natick, MA, USA). Because the platform’s movement was restricted to the antero-posterior direction, only 1D acceleration data along this axis were included in the dataset. Synchronization of the EEG and acceleration data, recorded by different devices, was achieved using transistor–transistor logic (TTL). Both the EEG and acceleration data were smoothed using the Savitzky–Golay filter (Matlab R2023b).

### 2.3. Data Analysis

To ensure uniformity across the subjects, detrending and baseline correction were applied to all event epochs. This was achieved by calculating the average and standard deviation of the EEG magnitude from 300 ms to 100 ms prior to the event onset and subtracting any linear trend or DC shifts. Subsequently, all EEG epochs within a channel were normalized by scaling them proportionally to the maximum EEG magnitude of the subject’s first event epoch, which was used as a reference point.

For this study, we focused on a single mastoid electrode site at R6 located at the center of the right mastoid bone. The EEG responses from each event were manually identified and categorized as either “Predicted” or “Unpredicted”. In total, 885 event EEGs were labeled as “Predicted”, while the remaining 435 were labeled as “Unpredicted”. Figure 2 illustrates the EEG data collected during two unpredictable events compared with ten predictable disturbances.

The onset of each event was defined as the initial acceleration of the dynamic platform and was determined using the platform accelerometer (Figure 3). We conducted experiments with different EEG epoch lengths, including 350 ms, 500 ms, 750 ms, and 1000 ms after the onset of the event, and evaluated the impact of selecting EEG epochs from 80 ms after the onset to the same cut-offs. EEG epochs were transferred into the state-space feature with 5th-order dynamic system state space matrices. The feature was then trained with the ensemble bagged trees classifier and 10-time 10-fold cross-validation.

### 2.4. State-Space Modeling

State-space-based system identification is a method that allows us to model and estimate the behavior of dynamic systems using measurement of the system’s input and output signals. It is important to note that unlike other systems, an EEG does not have any direct input signal; the EEG response reflects the cortical activity resulting from internal or external stimuli. Thus, we used an output-only system identification approach. This approach involves defining the state equation as
(1)$$x_{T+1}=Ax_{T}+{Ke}_{T}$$
where $x_{T}$ is the vector of internal states of the system at time T, A is the state matrix that defines the dynamic behavior of the system, K is the steady-state Kalman gain, and $e_{T}$ is the zero-mean white noise. ${Ke}_{T}$ together represent the process noise that affects the system’s internal states. The output equation is defined by
(2)$$y_{T}=Cx_{T}+e_{T}$$
where $y_{T}$ is the output EEG signal at time T and C is the output matrix that maps the internal states of the system to its output signals. The block diagram of (1) and (2) combined as a dynamic state-space model is shown in Figure 4. Finally, the estimation of the state matrices A, C, and K of the state-space dynamic system can be achieved using the n4sid method, which is described in detail in [32].
(3)$$A=\left[\begin{array}{ccc} a_{11} & \cdots & a_{1m}\\ \vdots & \ddots & \vdots \\ a_{m1} & \cdots & a_{mm} \end{array}\right],C=\left[\begin{array}{ccc} c_{1} & \ldots & c_{m} \end{array}\right],K={\left[\begin{array}{ccc} k_{1} & \ldots & k_{m} \end{array}\right]}^{T}$$

After estimating the state matrices A, C, and K, a feature vector can be created by concatenating each element of the matrices. In this case, a 5th-order state-space model was used, which resulted in a feature vector of 35 elements (1 × 35).
(4)$${feature}_{T}=\left[a_{11}\cdots a_{mm},c_{1}\cdots c_{m},k_{1}\cdots k_{m}\right]$$

### 2.5. Comparative Analysis

As previously mentioned, in our comparative analysis, we assessed the performance of our state-space method against a widely used EEG classification approach that incorporated AR coefficients and Shannon entropy features. Specifically, we utilized a 4th-order AR model combined with level 4 Shannon entropy values to construct a 1 × 20 feature vector that comprised 4 AR coefficients and 16 Shannon entropy values.

### 2.6. Real-Time Analysis

The real-time classifier was trained on data from 70% of the subjects (28 individuals), using 34 fall events labeled as “Expected” and “Unexpected”, along with 34 no-fall events labeled as “Expected”. For each event, EEG data (Ch. R6) and acceleration data were processed using a window size of 256 ms, with an 80 ms delay from the “onset” value. The selection of a short time window was driven by the necessity to predict falls within 400 ms, providing adequate time for communication, computation, and airbag deployment. The signals were filtered and normalized as previously described, and state-space features were extracted using a system order of 3 to facilitate rapid computation. The normalized EEG and acceleration threshold was defined as the minimum value among all maximum values of the “Unexpected” training data.

The algorithmic workflow for real-time fall detection is illustrated in Figure 5. A 256 ms window was initially selected at the onset of the EEG and acceleration signal recordings for each validation subject. During each iteration, the signals underwent normalization, followed by threshold evaluation. In cases where the signal fell below the defined threshold, the window advanced by 30 ms. Conversely, if the signal exceeded the threshold, state-space features were extracted and input into the classifier. When the classifier predicted an “Expected” outcome, the window advanced with a step size of 30 ms. However, if the result was classified as “Unexpected”, the window advanced by 1 s to mitigate the possibility of counting the same fall more than once. We note that the computational time for each iteration was estimated to be less than the step size (30 ms). A true positive or false positive classification was assigned for every event that exceeded the threshold based on the labeled data. In instances where a labeled event was not detected by the real-time algorithm, a false negative was recorded.

### 2.7. Performance Metrics

Sensitivity, specificity, and accuracy were the variables used as the indicators of performance. These measures are typically calculated from the number of true positive (TP), true negative (TN), false negative (FN), and false positive (FP) events predicted by the model, as determined by a “Standard of Truth” [33]. The sensitivity gauges the model’s ability to correctly identify instances of the positive class, while specificity measures its proficiency in recognizing the negative class. Accuracy provides an overall assessment of the model’s correctness by considering both true positives and true negatives in relation to all predicted outcomes. In this case, the “Unpredicted” class was considered the positive class and the “Predicted” class was considered the negative class.
(5)$$Sensitivity=\frac{TP}{TP+FN}$$
(6)$$Specificity=\frac{TN}{TN+FP}$$
(7)$$Accuracy=\frac{TP+TN}{TP+TN+FP+FN}$$

## 3. Results

Table 1 presents the classification performance of the AR model combined with the Shannon entropy features for various epoch lengths starting at the onset of the event (0 ms cut-off). The results indicate that as the epoch length increased, there was a notable improvement in the sensitivity, specificity, and accuracy. Specifically, the sensitivity increased from 60.3% for the 0–350 ms epoch to 74.3% for the 0–1000 ms epoch. Similarly, the specificity remained high, where it ranged from 90.1% to 94.3%, while the accuracy rose from 81.7% to 88.6%. Table 2 highlights the classification performance of the same AR and Shannon entropy features, but with epochs that started 80 ms after the onset of the event. In this scenario, the sensitivity showed an upward trend, beginning at 66.7% for the 80–350 ms epoch and reaching 76.8% for the 80–1000 ms epoch. The specificity remained high, ranging from 93.6% to 94.3%, while the accuracy improved from 86.0% to 89.3%.

Table 3 summarizes the performance of the system using the state-space method for different EEG epoch lengths starting at the onset of the event. The results indicate that as the epoch length increased, the overall accuracy of the state-space method also improved. Specifically, the accuracy improved from 93.1% for the 350 ms epoch to 94.5% for the 1000 ms epoch. The specificity of the system, which measured the proportion of true negatives correctly identified, remained consistently high across all epoch lengths, ranging from 96.1% to 96.9%. On the other hand, the sensitivity of the system, which measured the proportion of correctly identified true positives, increased as the epoch length increased. The sensitivity ranged from 87.1% for the 350 ms epoch to 90.9% for the 1000 ms epoch. Table 4 demonstrates that when the EEG epochs began at 80 ms after the onset of the event, the overall performance of the classification system improved across all epoch lengths. For example, the accuracy for the 80–1000 ms epoch length increased from 94.5% to 95.2%, while the sensitivity for the same epoch length rose from 90.9% to 91.9%. The specificity also showed improvement, with the highest value of 97.3% achieved for the 80–1000 ms epoch length.

The analysis thus far was conducted offline without leveraging the potential of the motion information. Recognizing that both signals contained valuable information about falls, we considered a comprehensive analysis that integrated both datasets. This section demonstrates that unexpected falls could be identified with an accuracy exceeding 99% by integrating the EEG and motion sensor data. The algorithm successfully detected 100% of the unexpected falls in over 85% of the subjects, with low rates of false positives and false negatives. Notably, the majority of errors were concentrated within a small subset of subjects. Table 5 summarizes the results across six training sets, demonstrating an average sensitivity of 93 ± 3%, specificity of 99.6 ± 0.1%, and accuracy of 99.5 ± 0.3%.

## 4. Discussion

In this study, we utilized a state-space based system identification approach to extract features from EEG signals for classifying postural perturbation events. We evaluated the impact of different epoch lengths on the classification performance, comparing the results from the event onset and 80 ms after the onset. The results indicate that the EEG signals following the postural perturbations were accurately classified, where longer epoch lengths generally improved the classification performance, though this improvement plateaued. Notably, using EEG epochs that started at 80 ms after the event onset yielded better performances than those that started at the onset.

The imbalance between “Predicted” and “Unpredicted” events in our dataset was an intentional aspect of the study design. In real-world scenarios, “Predicted” events dominate daily activities, but accuracy alone does not fully capture the detector performance. For example, a 97% accuracy might reflect high specificity but low sensitivity if the “Predicted” class makes up the majority of the data. To address this, we would increase the proportion of “Unpredicted” cases during training, aiming for a ratio of 1:2 or 1:3 between “Predicted” and “Unpredicted” classes. This will ensure that the classifier learns patterns from both classes, rather than focusing only on the dominant class, while still respecting the real-world prevalence. This approach emphasizes the importance of reliably detecting the rare yet critical “Unpredicted” events.

The superior classification performance of the state-space method stems from its ability to capture the dynamic nature of EEG signals by modeling temporal relationships within the data. In contrast, the AR and Shannon entropy methods primarily rely on static features, potentially overlooking complex temporal dynamics. The state-space approach generates a rich feature vector from state matrices that encapsulate both the dynamic behavior and statistical properties of EEG signals, enhancing its ability to discern patterns related to fall events. For example, for the 1000 ms epoch length, the state-space method achieved a sensitivity of 90.9% and an accuracy of 94.5%, while the AR and Shannon entropy methods reached only 74.3% sensitivity and 88.6% accuracy. This significant difference underscored the effectiveness of the state-space method in improving the classification performance for EEG-based fall detection.

Longer epoch lengths allow for capturing more temporal dynamics related to movement intention, providing additional data for training and testing the classifier. However, they also require longer wait times to collect data, which can delay fall detection and increase the computational time. In fall detection, speed is crucial, as delays can result in insufficient time for the protection system to deploy and prevent injury. Our results indicate that a 256 ms epoch can be sufficient when leveraging EEG and motion sensor fusion. Initiating EEG epochs at 80 ms following the onset of the event appears to further enhance the classifier’s performance. It is well documented that automatic postural reactions in the lower limb are elicited at approximately an 80–90 ms latency with a second peak between 90 and 120 ms [34,35]. Cortical EEG activity arising after 80 ms likely reflects the cortical component of the longer latency response [36], suggesting that EEG activity between 0 and 80 ms may not contain critical information for fall detection. Additionally, EEG signals in the 0–80 ms segment may be more vulnerable to noise or artifacts, potentially impacting classifier performance. Therefore, considering computational efficiency and detection speed, initiating epochs at 80 ms is preferable.

Despite these promising results, limitations existed in this study. The relatively small dataset may restrict the generalizability of our findings. A more significant and diverse dataset could help validate the results and improve the system performance across different populations, which could be achieved via collaborating with hospitals or research centers. A larger dataset would aid in refining the classifier parameters, adjusting the signal thresholds, and optimizing the training parameter set. This could improve the accuracy and reduce false events, which are the current barriers to the fall protection system. Furthermore, this study focused on one type of postural perturbation, which may not encompass the full range of real-life falls or daily activities. Future studies should evaluate the system’s effectiveness across various fall types and activities. Future reports will also include a more extensive analysis with results from additional EEG channels due to the fact our study only focused on the analysis of a single EEG channel.

Our current study focused on the fusion of EEG and accelerometer data to demonstrate the feasibility and potential of this approach for fall detection. In future work, we aim to expand upon this by integrating additional sensors, such as gyroscopes, pressure sensors, and visual sensors, to further enhance the accuracy and reliability of the system. The inclusion of multimodal data will provide a more comprehensive understanding of fall dynamics, contributing to the development of robust fall detection technologies that are effective, accessible, and capable of improving the safety and quality of life for the elderly and other vulnerable populations.

Refining the accuracy of our detection system requires exploring suitable features and algorithms for each data type and their synergistic integration. Advanced neural network architectures, such as Graph Neural Networks (GNNs), offer promising avenues for modeling complex relationships in neural data, as highlighted in [37]. Similarly, EEG-based analyses were applied to detect physiological states, including driver fatigue, using spatiotemporal fusion networks with brain-region-partitioning strategies [38]. Adopting such approaches could further enhance the classification performance in detecting perturbations.

Furthermore, while our proposed method achieved high accuracy in identifying unexpected falls, optimizing the algorithm to enhance real-time performance and reduce computational complexity is a crucial next step. These advancements are essential to translate our research findings into a practical fall detection system, which can be effectively deployed to enhance safety in elderly adults and vulnerable populations across various everyday environments.

## 5. Conclusions

Our study demonstrated the potential of utilizing EEG and motion data to detect postural perturbations and predict falls. The findings indicate that postural perturbations significantly modify EEG activity related to balance adjustments, which can be leveraged to differentiate between falls and non-falls. Notably, our state-space method outperformed the traditional autoregressive (AR) and Shannon entropy (SE) methods, where it achieved a sensitivity of 90.9%, specificity of 97.3%, and accuracy of 95.2%. By integrating the EEG and motion sensor data, we demonstrated that unexpected falls could be detected in real time with over 99% accuracy. Future research should focus on utilizing larger and more diverse datasets, investigating the system’s performance across various types of falls and daily living activities, and expanding the analysis to include EEG data from multiple channels while integrating other sensors, such as accelerometers. Developing real-time processing and classification methods will further enhance the effectiveness of fall detection systems. Overall, this study underscored the promising role of EEG data, particularly when combined with other sensor modalities, in creating more accurate and efficient fall detection systems that can improve the safety and quality of life for elderly adults and individuals with mobility impairments.

## Figures and Tables

**Figure 1 sensors-24-07779-f001:**
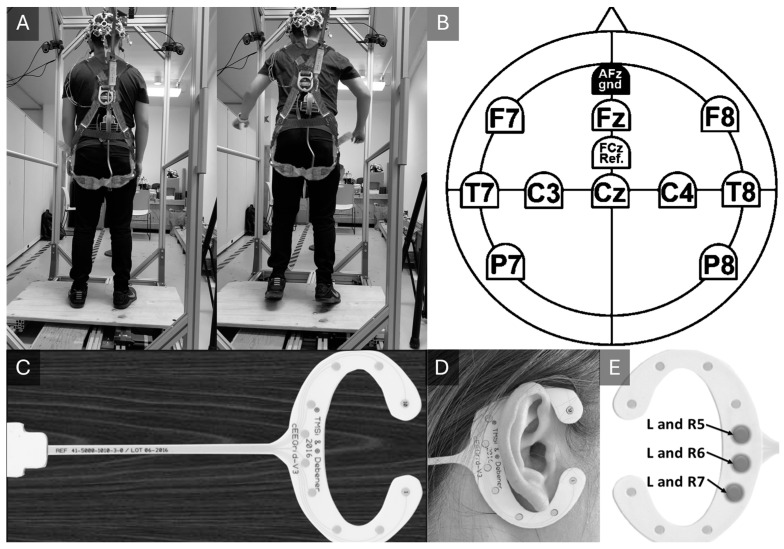
(**A**) Experimental setup illustration, showing “upright stance” (**left**) and “balance perturbation” (**right**) stages. (**B**) Scalp EEG channel locations. (**C**) Full view of the cEEGrid piece. (**D**) Illustration of subject wearing the cEEGrids. (**E**) Mastoid channel locations.

**Figure 2 sensors-24-07779-f002:**
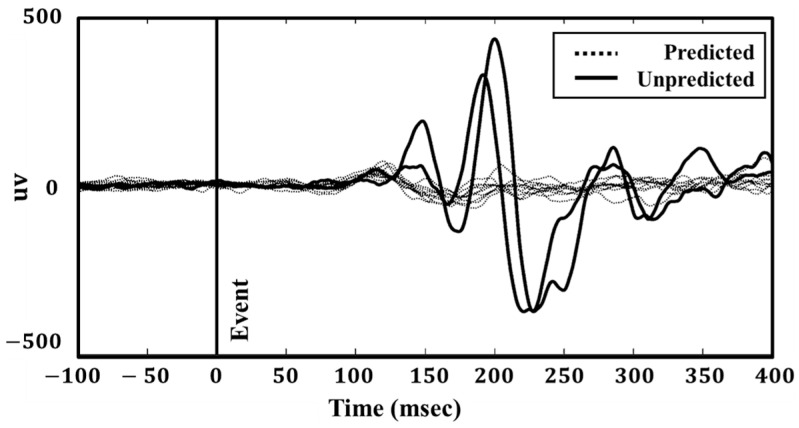
Exemplary EEG (Ch. R6) activity during two unpredictable events and ten predictable events. Minimal differences in the EEG activity were observed between these two types of events up to 80 ms following the event onset. The vertical line marks the onset of each event.

**Figure 3 sensors-24-07779-f003:**
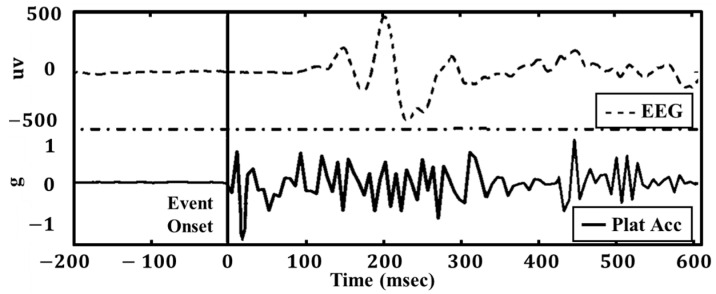
Exemplar event data illustrating EEG (Ch. R6) and platform-based acceleration. The EEG data shown were detrended, baseline corrected, and filtered using a 2nd-order Butterworth bandpass filter (2.5 Hz–30 Hz) and a 60 Hz notch filter. The acceleration data represents one-dimensional raw measurements in the antero-posterior direction from the platform. The vertical line marks the onset of the event.

**Figure 4 sensors-24-07779-f004:**
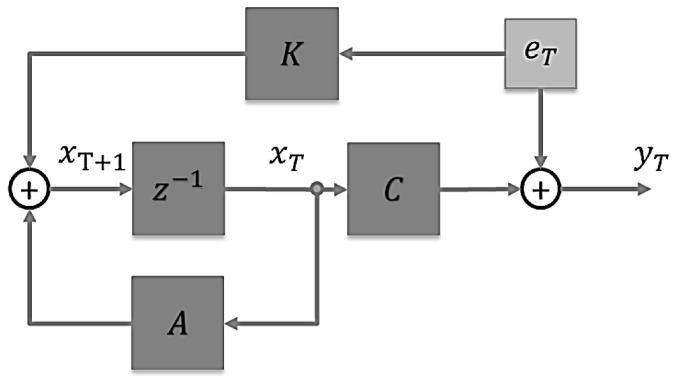
Block diagram for the proposed dynamic state-space model.

**Figure 5 sensors-24-07779-f005:**
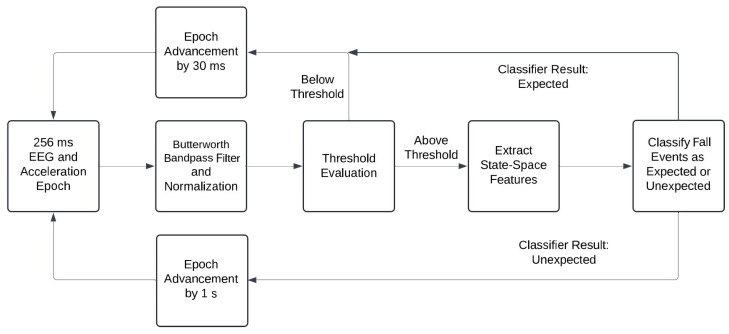
Algorithm workflow for real-time fall detection using EEG and acceleration signals.

**Table 1 sensors-24-07779-t001:** Results of classification for different windows (0 ms–cut-off) using AR and SE features.

Epoch Length	Sensitivity	Specificity	Accuracy
0–350 ms	60.3%	90.1%	81.7%
0–500 ms	62.9%	91.6%	83.5%
0–750 ms	72.2%	94.8%	88.4%
0–1000 ms	74.3%	94.3%	88.6%

**Table 2 sensors-24-07779-t002:** Results of classification for different windows (80 ms–cut-off) using AR and SE features.

Epoch Length	Sensitivity	Specificity	Accuracy
80–350 ms	66.7%	93.6%	86.0%
80–500 ms	69.2%	93.5%	86.6%
80–750 ms	73.4%	94.0%	88.1%
80–1000 ms	76.8%	94.3%	89.3%

**Table 3 sensors-24-07779-t003:** Results of classification for different windows (0 ms–cut-off) using state-space features.

Epoch Length	Sensitivity	Specificity	Accuracy
0–350 ms	87.1%	96.1%	93.1%
0–500 ms	88.7%	96.6%	94.0%
0–750 ms	89.1%	96.8%	94.1%
0–1000 ms	90.9%	96.9%	94.5%

**Table 4 sensors-24-07779-t004:** Results of classification for different windows (80 ms–cut-off) using state-space features.

Epoch Length	Sensitivity	Specificity	Accuracy
80–350 ms	87.9%	96.7%	93.9%
80–500 ms	89.1%	97.0%	94.4%
80–750 ms	89.4%	97.2%	94.6%
80–1000 ms	90.9%	97.3%	95.2%

**Table 5 sensors-24-07779-t005:** Results of real-time sensor fusion algorithm.

Training Set	Sensitivity	Specificity	Accuracy
1	98.1%	99.7%	99.7%
2	94.7%	99.6%	99.6%
3	90.1%	99.6%	99.5%
4	87.5%	99.60%	98.9%
5	95.3%	99.8%	99.8%
6	93.1%	99.5%	99.5%
Average	93 ± 3%	99.6 ± 0.1%	99.5 ± 0.3%

## Data Availability

The data supporting this study’s findings are available from the corresponding authors upon reasonable request. The data is not publicly available due to the data sharing restriction from the IRB.

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
