# Peer review of "Real-Time Postural Disturbance Detection Through Sensor Fusion of EEG and Motion Data Using Machine Learning"

_sensors, 2024, doi:10.3390/s24237779_

Round 1
Reviewer 1 Report
Comments and Suggestions for Authors
Your research investigates the use of EEG data and machine learning techniques to develop a real-time fall detection system.The study introduces a state-space model-based method for analyzing EEG signals, demonstrating its superior performance compared to other feature extraction approaches.Additionally, a real-time algorithm integrating EEG and accelerometer data achieves accurate fall detection within 400 milliseconds, with over 99% accuracy for unexpected falls.These findings suggest that EEG data can significantly enhance the accuracy and efficiency of fall detection systems, offering a promising avenue for developing safer fall detection technologies.
The topic is interesting, but there are some major concerns that that need to be addressed:
-
Dataset Diversity: Although this paper uses data from young and elderly adults, the sample size is relatively small. It is recommended to expand the size of the dataset and consider subjects of different races, genders, and health conditions to improve the generalizability of the model.
-
Diversity of Fall Types: This paper focuses on one type of fall event, so it is recommended to consider different types of fall events in future research, such as falls, slips, and trips, as well as daily living activities, to improve the robustness of the model.
-
Utilization of Multi-channel EEG Data: This paper only uses a single EEG channel, so it is recommended to utilize multi-channel EEG data in future research and explore more effective feature extraction and classification methods.
-
Fusion with Other Sensors: This paper only uses EEG and accelerometer data, so it is recommended to consider fusion with data from other sensors in future research, such as gyroscopes, pressure sensors, and visual sensors, to improve the accuracy and reliability of the model.
-
Optimization of Real-time Performance: Although this paper’s proposed method achieved a high accuracy in identifying unexpected falls, it is recommended to further optimize the algorithm in future research to improve real-time performance and reduce computational complexity.
- Recommend citing the following papers:
â‘ Z. Wu, S. Pan, F. Chen, G. Long, C. Zhang and P. S. Yu, "A Comprehensive Survey on Graph Neural Networks," in IEEE Transactions on Neural Networks and Learning Systems, vol. 32, no. 1, pp. 4-24, Jan. 2021, doi: 10.1109/TNNLS.2020.2978386.
â‘¡Hu F, Zhang L, Yang X, et al. EEG-Based Driver Fatigue Detection Using Spatio-Temporal Fusion Network With Brain Region Partitioning Strategy[J]. IEEE Transactions on Intelligent Transportation Systems, 2024, 25(8): 9618-9630.
The two papers mentioned above are some of the more cutting-edge and outstanding papers in the field of artificial intelligence in recent years. Citing them can enhance the completeness of the article.Highlight that their study can be enriched by referencing the paper titled "EEG-Based Driver Fatigue Detection Using Spatio-Temporal Fusion Network With Brain Region Partitioning Strategy.
â‘¢H. Liang, Y. Wang, H. Li, Y. Wang, P. X. Liu and R. Liu, "Development and Characterization of a Dry Ear-EEG Sensor With a Generic Flexible Earpiece," in IEEE Transactions on Instrumentation and Measurement, vol. 72, pp. 1-12, 2023, Art no. 4006212, doi: 10.1109/TIM.2023.3277949.
Citing this article demonstrates the potential application of the novel ear-EEG sensor in the field of brain-computer interfaces, providing a reference for developing more convenient, cost-effective, and efficient BCI devices.
Comments on the Quality of English Language-
Pay attention to the use of long sentences: Some sentences in the article are quite long, and can be broken down into shorter sentences to improve readability.
-
Explain technical terms: For some less common technical terms, consider explaining them when they first appear to help readers understand.
Author Response
Comment 1:
Your research investigates the use of EEG data and machine learning techniques to develop a real-time fall detection system. The study introduces a state-space model-based method for analyzing EEG signals, demonstrating its superior performance compared to other feature extraction approaches. Additionally, a real-time algorithm integrating EEG and accelerometer data achieves accurate fall detection within 400 milliseconds, with over 99% accuracy for unexpected falls. These findings suggest that EEG data can significantly enhance the accuracy and efficiency of fall detection systems, offering a promising avenue for developing safer fall detection technologies.
The topic is interesting, but there are some major concerns that need to be addressed:
Response 1:
Thank you for your thoughtful and detailed comments. We appreciate your recognition of our work and your insights into its potential impact. Your feedback is both encouraging and invaluable for improving our research. We have carefully addressed your concerns, and our detailed responses are provided below.
Comment 2:
Dataset Diversity: Although this paper uses data from young and elderly adults, the sample size is relatively small. It is recommended to expand the size of the dataset and consider subjects of different races, genders, and health conditions to improve the generalizability of the model.
Response 2:
We acknowledge the relatively small sample size in our phase 1 study, which we have discussed in the fifth paragraph of the Discussion. The primary goal of this paper is to present preliminary findings demonstrating the feasibility and potential of our approach, which is crucial for securing phase 2 funding to recruit a larger and more diverse set of subjects. [page 10, line 353-359]
Our current dataset includes 10 older males, 10 older females, 10 younger males, and 10 younger females, ensuring a balanced representation of both age and gender. We have also updated the Methods section under the "Participants" subsection to include detailed inclusion and exclusion criteria for clarity. In the next phase, we aim to expand the dataset significantly, incorporating participants of diverse races, genders, and health conditions to enhance the model's generalizability. [page 3, line 130-144]
Comment 3:
Diversity of Fall Types: This paper focuses on one type of fall event, so it is recommended to consider different types of fall events in future research, such as falls, slips, and trips, as well as daily living activities, to improve the robustness of the model.
Response 3:
We agree with your suggestion and have already addressed this in the second half of paragraph 5 in the Discussion. Expanding the diversity of fall types, including slips, trips, and daily living activities, is part of our plan for the phase 2 study to improve the model's robustness. [page 10, line 359-362]
Comment 4:
Utilization of Multi-channel EEG Data: This paper only uses a single EEG channel, so it is recommended to utilize multi-channel EEG data in future research and explore more effective feature extraction and classification methods.
Response 4:
We agree with your recommendation and have addressed this issue at the end of paragraph 4 in the Discussion. Our current study focused on a single EEG channel based on findings from our previous work using the same dataset, " Wang, Z., Graci, V., Seacrist, T., Guez, A., & Keshner, E. A. (2023). Localizing EEG recordings associated with a balance threat during unexpected postural translations in young and elderly adults. IEEE Transactions on Neural Systems and Rehabilitation Engineering, 31, 4514-4520.," which demonstrated that perturbation-evoked potentials (PEP) can be reliably identified using a single EEG sensor placed over the mastoid bone. Future devices should be designed to be wearable and ergonomic. To support this, we demonstrate that predicting falls using only a single EEG channel is feasible.
However, we do agree that further research utilizing multi-channel EEG data is essential to provide a more comprehensive understanding and improve feature extraction and classification methods, such a statement can also be found between lines 362 and 364 on page 10.
Comment 5:
Fusion with Other Sensors: This paper only uses EEG and accelerometer data, so it is recommended to consider fusion with data from other sensors in future research, such as gyroscopes, pressure sensors, and visual sensors, to improve the accuracy and reliability of the model.
Response 5:
We acknowledge the potential of integrating additional sensors such as gyroscopes, pressure sensors, and visual sensors in future studies. We plan to include these in our research as part of our ultimate goal to support the development of fall detection technologies that are both effective and accessible. We have accordingly added the 6th paragraph of the Discussion to emphasize this point. [page 10, line 365-372]
Comment 6:
Optimization of Real-time Performance: Although this paper’s proposed method achieved a high accuracy in identifying unexpected falls, it is recommended to further optimize the algorithm in future research to improve real-time performance and reduce computational complexity.
Response 6:
We agree that further optimization of the algorithm to enhance real-time performance and reduce computational complexity is necessary. This aligns with our long-term goal of developing a practical and efficient fall detection system. We have modified the last two paragraphs of the discussion in the revised manuscript to emphasize its importance and outline our plans for future research. [page 10, line 373-386]
Comment 7:
Recommend citing the following papers:
- Wu, S. Pan, F. Chen, G. Long, C. Zhang and P. S. Yu, "A Comprehensive Survey on Graph Neural Networks," in IEEE Transactions on Neural Networks and Learning Systems, vol. 32, no. 1, pp. 4-24, Jan. 2021, doi: 10.1109/TNNLS.2020.2978386.
- Hu F, Zhang L, Yang X, et al. EEG-Based Driver Fatigue Detection Using Spatio-Temporal Fusion Network With Brain Region Partitioning Strategy[J]. IEEE Transactions on Intelligent Transportation Systems, 2024, 25(8): 9618-9630.
The two papers mentioned above are some of the more cutting-edge and outstanding papers in the field of artificial intelligence in recent years. Citing them can enhance the completeness of the article. Highlight that their study can be enriched by referencing the paper titled "EEG-Based Driver Fatigue Detection Using Spatio-Temporal Fusion Network With Brain Region Partitioning Strategy.
- Liang, Y. Wang, H. Li, Y. Wang, P. X. Liu and R. Liu, "Development and Characterization of a Dry Ear-EEG Sensor With a Generic Flexible Earpiece," in IEEE Transactions on Instrumentation and Measurement, vol. 72, pp. 1-12, 2023, Art no. 4006212, doi: 10.1109/TIM.2023.3277949.
Citing this article demonstrates the potential application of the novel ear-EEG sensor in the field of brain-computer interfaces, providing a reference for developing more convenient, cost-effective, and efficient BCI devices.
Response 7:
Thank you for your suggestion. We have incorporated the recommended citations into the manuscript. Specifically:
1 as [37] and 2 as [38] in discussion [page 10, line 376 and line 378] and 3 as [24] in introduction [page 2, line 91-92].
Reviewer 2 Report
Comments and Suggestions for Authors
Abstract
I understand that writing a concise abstract can be challenging; however, the current version lacks sufficient background information. Aside from the sentence: “Millions of people around the globe are impacted by falls annually, making it a significant public health concern,” there is little context provided. It would be valuable to include an additional sentence to further justify the approach and aim of the study.
References
The reference format may not align with the journal's guidelines. Please review the submission guidelines carefully to ensure compliance.
Introduction
The description of the previous study is engaging and informative; however, it is not necessary to describe the full methodology in the Introduction. I recommend reducing the level of detail, particularly in lines 92–106.
While it is crucial to explain the connection between this study and your previous work, detailed methodology should be confined to the Methods section (lines 107–120). Including this information in both sections creates redundancy.
Similarly, in lines 123–130, the discussion of methodological details (following Objective 1) should also be moved to the Methods section to avoid repetition.
Methods
The methodology was fascinating, particularly the balance perturbation procedure and algorithm development. To enhance clarity and organization, consider introducing subheadings such as:
- Population/Enrollment
- Experimental Procedure
- EEG Procedure
- Analysis
Figures
Regarding “Figure 1. (A) Illustration of experimental setup,” the figure does not clearly depict the balance procedure. Adding more figures or illustrations showing different stages of the balance procedure would improve understanding.
Additionally, avoid using the "XD" emoji to obscure the face in the figure. It would be more professional to cover the face with a circle or other neutral graphic.
Participant Information
There is little demographic information provided about the participants beyond the total number. Including details such as age, sex, education, or any clinical and demographic characteristics collected would be crucial.
Have you considered analyzing differences between the two groups (e.g., young vs. elderly participants)? This could help identify whether specific factors, such as age, affect the results.
Event EEG Labeling
In the methodology, you note that 885 EEG events were labeled as "Predicted," while 435 were labeled as "Unpredicted." Could this imbalance between the two groups influence your results? A brief discussion of this potential effect would strengthen the interpretation.
Figure 2 and Statistical Comparisons
If I understand correctly, Figure 2 illustrates that perturbation-evoked potentials (PEP) differ between Predictable and Unpredictable situations. Was any statistical comparison performed on these results? Additionally, did you compare PEPs between the “Young” and “Elderly” groups in both Predictable and Unpredictable conditions? I would expect the differences to diminish in the elderly, potentially explaining their higher rates of unexpected falls.
Discussion
The discussion could benefit from further development. While the results are well-summarized, it is equally important to compare your findings to existing literature and provide suggestions for future studies. Additionally, consider discussing how your results could be applied in real-life scenarios or monitoring solutions.
Currently, the discussion lacks sufficient comparisons to previous research and future recommendations, which would strengthen the overall impact of your study. On the other hand, the limitations are clearly described and well-addressed.
Author Response
Comment 1:
Abstract
I understand that writing a concise abstract can be challenging; however, the current version lacks sufficient background information. Aside from the sentence: “Millions of people around the globe are impacted by falls annually, making it a significant public health concern,” there is little context provided. It would be valuable to include an additional sentence to further justify the approach and aim of the study.
Response 1:
We appreciate your feedback and agree that adding more background information to the abstract will provide better context for readers. We have revised the abstract by adding a sentence [page 1, line 16-17] to justify the approach and aim of the study.
Comment 2:
References
The reference format may not align with the journal's guidelines. Please review the submission guidelines carefully to ensure compliance.
Response 2:
We apologize for this oversight. The references are now updated in ACS style.
Comment 3:
Introduction
The description of the previous study is engaging and informative; however, it is not necessary to describe the full methodology in the Introduction. I recommend reducing the level of detail, particularly in lines 92–106.
While it is crucial to explain the connection between this study and your previous work, detailed methodology should be confined to the Methods section (lines 107–120). Including this information in both sections creates redundancy.
Similarly, in lines 123–130, the discussion of methodological details (following Objective 1) should also be moved to the Methods section to avoid repetition.
Response 3:
We agree that the level of detail in the Introduction regarding the methodology could be reduced. The revised manuscript now ensures that the Introduction provides sufficient context without repeating methodological specifics already covered in the Methods section. [Last four paragraphs in Introduction, page 3, line 96-128]
Comment 4:
Methods
The methodology was fascinating, particularly the balance perturbation procedure and algorithm development. To enhance clarity and organization, consider introducing subheadings such as:
Population/Enrollment
Experimental Procedure
EEG Procedure
Analysis
Response 4:
We have now added subheadings to enhance clarity and organization in the Method section.
Comment 5:
Figures
Regarding “Figure 1. (A) Illustration of experimental setup,” the figure does not clearly depict the balance procedure. Adding more figures or illustrations showing different stages of the balance procedure would improve understanding.
Additionally, avoid using the "XD" emoji to obscure the face in the figure. It would be more professional to cover the face with a circle or other neutral graphic.
Response 5:
We have revised Figure 1 to improve clarity. Panel (A) now includes illustrations showing both "upright stance" and "balance perturbation" stages of the procedure to better depict the process. Additionally, the "XD" emoji has been removed. [page 4]
Comment 6:
Participant Information
There is little demographic information provided about the participants beyond the total number. Including details such as age, sex, education, or any clinical and demographic characteristics collected would be crucial.
Response 6:
Our current dataset includes 10 older males, 10 older females, 10 younger males, and 10 younger females, ensuring a balanced representation of both age and gender. We have also updated the Methods section under the "Participants" subsection to include detailed inclusion and exclusion criteria for clarity. [page 3, line 130-144]
Comment 7:
Have you considered analyzing differences between the two groups (e.g., young vs. elderly participants)? This could help identify whether specific factors, such as age, affect the results.
Response 7:
We have considered analyzing differences among different groups. In our previous work, "Wang, Z., Graci, V., Seacrist, T., Guez, A., & Keshner, E. A. (2023). Localizing EEG recordings associated with a balance threat during unexpected postural translations in young and elderly adults. IEEE transactions on neural systems and rehabilitation engineering, 31, 4514-4520.," we explored such differences from a statistical perspective. For future research, we plan a more comprehensive comparison using predictive modeling approaches, with an expanded dataset that includes diverse subjects across age, race, gender, and health conditions to improve the generalizability of the model.
Comment 8:
Event EEG Labeling
In the methodology, you note that 885 EEG events were labeled as "Predicted," while 435 were labeled as "Unpredicted." Could this imbalance between the two groups influence your results? A brief discussion of this potential effect would strengthen the interpretation.
Response 8:
We do not believe the imbalance between 'Predicted' and 'Unpredicted' groups poses a significant issue. Our protocol was intentionally designed so that most perturbations would be labeled as 'Predicted.' While detecting both classes is important, the 'Unpredicted' class holds greater practical significance, as it represents fall accidents that could lead to severe injury or death. Our primary focus is to ensure high performance in detecting 'Unpredicted' events, which are critical for real-world applications.
In practical scenarios, the 'Predicted' class would naturally dominate, accounting for over 99% of daily activities. However, accuracy alone cannot fully represent the detector's actual performance. For instance, a 90% accuracy might result from 99% specificity and only 50% sensitivity if 80% of the dataset belongs to the 'Predicted' class. To address this, we intentionally increased the ratio of 'Unpredicted' cases during training, aiming to maintain a ratio of approximately 1:2 or 1:3 between 'Predicted' and 'Unpredicted' classes. For instance, even in a dataset where 'Predicted' events significantly outnumber 'Unpredicted' events, we would still randomly exclude some 'Predicted' cases to achieve the desired ratio.
This strategy ensures the classifier has a fair opportunity to learn patterns from both classes, rather than focusing disproportionately on the dominant class, while still respecting the real-world prevalence of 'Predicted' cases. This balance highlights the importance of a system capable of reliably identifying the rare but critical 'Unpredicted' events.
We have added a paragraph in the Discussion section to stress this point. [page 9, line 317-326]
Comment 9:
Figure 2 and Statistical Comparisons
If I understand correctly, Figure 2 illustrates that perturbation-evoked potentials (PEP) differ between Predictable and Unpredictable situations. Was any statistical comparison performed on these results? Additionally, did you compare PEPs between the “Young” and “Elderly” groups in both Predictable and Unpredictable conditions? I would expect the differences to diminish in the elderly, potentially explaining their higher rates of unexpected falls.
Response 9:
Statistical comparisons on perturbation-evoked potentials (PEPs) between Predictable and Unpredictable situations were conducted in our previous work: “Wang, Z., Graci, V., Seacrist, T., Guez, A., & Keshner, E. A. (2023). Localizing EEG recordings associated with a balance threat during unexpected postural translations in young and elderly adults. IEEE Transactions on Neural Systems and Rehabilitation Engineering, 31, 4514-4520”. This prior study also included comparisons of PEPs across age groups (Young vs. Elderly) and between sexes under both conditions. The findings revealed age- and sex-related differences, with diminished responses in the elderly potentially contributing to their higher rates of unexpected falls.
For this manuscript, we have focused on classifying Predictable and Unpredictable events using EEG and have deferred additional age- and sex-related analyses to future work.
Comment 10:
Discussion
The discussion could benefit from further development. While the results are well-summarized, it is equally important to compare your findings to existing literature and provide suggestions for future studies. Additionally, consider discussing how your results could be applied in real-life scenarios or monitoring solutions.
Currently, the discussion lacks sufficient comparisons to previous research and future recommendations, which would strengthen the overall impact of your study. On the other hand, the limitations are clearly described and well addressed.
Response 10:
We have excessively revised the Discussion to address the issues mentioned:
- Comparison to Existing Literature: We included a comparison of our state-space approach to AR modeling and Shannon entropy methods, emphasizing its advantages in capturing EEG signal dynamics. In addition, we referenced advancements in neural network architectures like Graph Neural Networks [37] and spatiotemporal fusion networks [38] to contextualize our study within current research.
- Future Recommendations: We added plans for future work, such as expanding the dataset for better generalizability, integrating additional sensors for multimodal data, and optimizing for real-time performance. We also briefly explained the rationale behind the imbalance of our dataset class and how it mimics and could affect the real-world scenarios.
Round 2
Reviewer 1 Report
Comments and Suggestions for Authors
I would like to thank the authors for their significant efforts in revising the manuscript and addressing my concerns. All responses are satisfactory, and I have no further comments on the revised manuscript.
Comments on the Quality of English LanguageN/A